# Using Tangible User Interfaces (TUIs): Preliminary Evidence on Memory and Comprehension Skills in Children with Autism Spectrum Disorder

**DOI:** 10.3390/bs15030267

**Published:** 2025-02-25

**Authors:** Mariagiovanna De Luca, Ciro Rosario Ilardi, Pasquale Dolce, Angelo Rega, Raffaele Di Fuccio, Franco Rubinacci, Maria Gallucci, Paola Marangolo

**Affiliations:** 1Department of Humanities Studies, University Federico II, Via Porta di Massa 1, 80133 Naples, Italy; mariagiovannadeluca1997@gmail.com; 2Interdepartmental Research Centre on Management and Innovation in Healthcare (CIRMIS), University Federico II, Via Pansini 5, 80131 Naples, Italy; cirorosario.ilardi@gmail.com; 3Dipartimento di Scienze Mediche Translazionali, University Federico II, Corso Umberto I 40, 80138 Naples, Italy; pasquale.dolce@unina.it; 4Neapolisanit Rehabilitation Centre, Via Funari, 80044 Naples, Italy; angelo.rega@unipegaso.it (A.R.); m.gallucci@neapolisanit.net (M.G.); 5Department of Psychology and Educational Sciences, Telematic University of Pegaso, Piazza dei Santi Apostoli 49, 00187 Rome, Italy; raffaele.difuccio@unipegaso.it; 6Smarted srl, Riviera di Chiaia, 256, 80121 Naples, Italy; franco.rubinacci@smarted.it

**Keywords:** autism spectrum disorder, neurodevelopmental disorders, tangible user interfaces, multisensory integration, storytelling

## Abstract

Autism spectrum disorder (ASD) is a complex neurodevelopmental condition involving persistent challenges with social communication, as well as memory and language comprehension difficulties. This study investigated the effects of a storytelling paradigm on language comprehension and memory skills in children with ASD. A traditional approach, using an illustrated book to deliver the narrative, was compared to a novel paradigm based on Tangible User Interfaces (TUIs) combined with multisensory stimulation. A group of 28 children (ages between 6 and 10 years old) was asked to listen to a story over four weeks, two times a week, in two different experimental conditions. The experimental group (*n* = 14) engaged with the story using TUIs, while the control group (*n* = 14) interacted with a corresponding illustrated book. Pre- and post-intervention assessments were conducted using NEPSY-II subtests on language comprehension and memory. At the end of the intervention, a trend of improved performance was found. In particular, a greater number of subjects benefited from the intervention in the experimental group compared with the control group in instruction comprehension and narrative memory-cued recall. These preliminary findings suggest that TUIs may enhance learning outcomes for children with ASD, warranting further investigation into their potential benefits.

## 1. Introduction

### 1.1. Autism Spectrum Disorder (ASD)

Autism spectrum disorder (ASD) is a complex neurodevelopmental disorder marked by difficulties in social interaction and communication and the presence of restricted or repetitive behavior, all of which can greatly affect daily life and overall well-being ([37]). Early diagnosis and intervention are essential in providing tailored support for individuals with ASD and optimizing their developmental outcomes. Although the exact cause of ASD remains unclear, research indicates that genetic factors are influential, with specific gene variations and mutations linked to an increased likelihood of ASD ([54]). Additionally, non-genetic influences, such as prenatal exposure to environmental toxins, maternal health during pregnancy, and early life experiences, may also play a role in the development of this profile ([44]; [49]). 

The term “spectrum” in ASD reflects the wide range of symptoms and varying levels of severity experienced by ASD individuals. According to the DSM-5, ASD is categorized into three levels based on the severity of symptoms and the degree of support required ([31]; [55]). Level 1 (Mild ASD), formerly referred to as high-functioning autism, includes those individuals who may face challenges in social communication and interaction. They often display restricted or repetitive behaviors but might also excel in specific areas, such as language and cognitive abilities. While less support is needed than in more severe forms, tailored interventions and assistance could enhance their quality of life. Level 2 (Moderate ASD) individuals experience more pronounced difficulties with social communication and often find it harder to “mask” or blend into social situations. Challenges with understanding social norms and maintaining interactions are common. They may also display repetitive behaviors, strong adherence to routines, and intense interests that stand out from neurotypical behavior. Moderate support is typically required to navigate daily life and social situations. Level 3 (Severe ASD) individuals, often referred to as low-functioning autism, have significant impairments in communication and social interaction. Self-regulation is more challenging, and they are at higher risk for neglect or discrimination. Communication, both verbal and non-verbal, can be difficult, and they may engage in pronounced repetitive behaviors. Social interactions are limited, and people often prefer solitary activities due to difficulty in understanding social cues and norms. Intensive support is necessary to help them engage meaningfully in daily life ([37]; [79]).

Several studies have shown that ASD individuals often experience difficulties in comprehending task instructions, with a general impairment in narrative comprehension ([12]; [48]). In a storytelling task, they may find it difficult to understand the structure and details of the story ([12]; [48]; [80]). Still, other studies have highlighted that individuals with ASD—whether high- or low-functioning—often exhibit deficits in verbal memory tasks, particularly when recalling word lists ([8]; [10]; [17]; [88]). These memory troubles become more evident when support cues are absent.

### 1.2. Traditional Interventions for Autism Spectrum Disorder

Occupational therapy and Applied Behavior Analysis (ABA) are commonly used interventions for ASD that are aimed at empowering academic and functional skills, as well as visuomotor abilities ([30]). However, conventional occupational approaches, such as shape-matching training, often struggle to sustain engagement and adequately address the complex needs of individuals with ASD ([47]). Still, while ABA-driven interventions have shown promise, their effectiveness remains debated due to methodological issues and limited generalization to real-life contexts ([92]). Furthermore, the high intervention intensity (up to 40 h per week) poses challenges for both children and their families ([45]). Also, ethical concerns have been raised by autism rights and neurodiversity activists ([45]), and uncertainty remains regarding potential conflicts of interest among researchers within the ABA industry ([7]).

### 1.3. The Role of Technology

A growing body of research has highlighted the significant advantages of integrating computer technologies into autism rehabilitation, especially in fostering motivation, enhancing attentional focus, minimizing maladaptive behaviors, and improving learning outcomes. Furthermore, technology-based interventions provide consistency, predictability, and immediate feedback while establishing clear routines and expectations, all of which satisfy the needs of children with ASD ([73]).

Overall, the most widely used technologies in ASD rehabilitation can be broadly classified into three domains: standard computer systems, robotic technology, and virtual reality. Nevertheless, the use of standard computer systems poses challenges for individuals with limited dexterity and fine motor skills ([74]). As for robotics and virtual reality, both approaches have shown promise in stimulating social skills and motivation ([21]; [47]); however, they may be less effective in improving cognitive functioning. 

Recent studies have emphasized the effectiveness of interventions that leverage multisensory integration (MSI) for individuals facing memory and language comprehension deficits ([62]). Human communication typically relies on multiple sensory channels—such as facial expressions, affective vocalizations and touch—rather than solely on unimodal signal transmission. MSI is thus essential for enhancing both communicative and social behaviors ([62]). Among the earliest senses to develop, the human sense of smell retains critical social communicative functions ([43]; [59]; [78]). Furthermore, the interplay between auditory and visual modalities is fundamental for processing speech stimuli ([77]; [83]; [85]). Previous neuroimaging meta-analyses have identified a neural network involved in multimodal sensory input processing, which includes fronto-temporal and limbic regions. These brain structures support not only the strictly linguistic aspects of verbal communication but also its connotative features, such as affective and emotional tones ([6]; [71]). 

Some MSI-based interventions have been specifically designed for individuals with ASD. Among these, the “Snoezelen” approach ([41]) involves the simultaneous stimulation of multiple senses with the aim of reducing stereotyped behaviors ([41]). For children with ASD and limited verbal abilities, Parés and coworkers introduced “MEDIATE,” an interactive environment that generates real-time visual and auditory stimuli to promote creativity, exploration, and enjoyment ([61]). Existing research primarily focused on vision and hearing, largely overlooking the senses of touch and smell ([3]; [43]; [59]).

### 1.4. Tangible User Interfaces (TUIs) as a Novel Approach

In these last years, Tangible User Interfaces (TUIs) have emerged as a groundbreaking methodology. TUIs are physical interfaces that introduce a new way of “grasping” computer technology. Specifically, they allow users to engage with a computer system through a combination of physical/perceptual and digital actions. An intriguing application of TUIs is in the context of storytelling tasks within a mixed-reality environment ([40]). In this scenario, TUIs facilitate user interaction by bridging real objects and digital stimuli within a given narrative framework. As TUIs support multisensory stimulation (e.g., visual, auditory, olfactory feedback), a TUI-mediated storytelling paradigm promoting MSI is expected to boost memory and language comprehension abilities ([24]; [63]). Interestingly, research involving typically developing or visually impaired children has shown that olfactory stimulation can positively influence text memorization ([34]; [75]). 

An advantage of TUIs over standard technology protocols lies in their ability to harness physical object manipulation. Unlike traditional computer-based methods relying on symbolic or graphical representations, TUIs streamline human–computer interaction by integrating tangible objects. This feature decreases the cognitive and fine motor demands typically associated with tasks such as drag-and-drop operations using a mouse ([29]). When applied to storytelling tasks, TUIs can incorporate MSI into the learning process, likely producing greater engagement and improving cognitive outcomes. Moreover, by building semi-ecological learning environments that closely resemble real-world scenarios, TUIs may enhance generalizability. Indeed, children may find it easier to apply the skills they learned during training sessions to everyday life ([91]). 

In previous research, TUIs have been mainly employed in educational settings ([22]; [24]; [63]) to foster social competencies ([1]; [32]; [84]) and improve emotion recognition ([65]). Conversely, little is known about their impact on cognitive functioning ([1]; [5]; [28]; [74]). Applying a similar paradigm within a structured cognitive rehabilitation program may be highly beneficial. Similarly, multisensory stimulation, which activates brain networks essential for both verbal and non-verbal communication ([2]; [71]), may prove effective in improving cognitive skills in individuals with ASD ([67]). 

Building on these insights, the present study aimed at evaluating the effectiveness of TUIs, integrated with a multisensory approach, in improving verbal memory and instruction comprehension in children with ASD engaged in a storytelling task. These domains are significantly impaired in children with ASD ([8]; [48]). Addressing these deficits might enhance the effectiveness of any subsequent rehabilitation interventions. The TUI-mediated intervention was compared to a traditional approach, where the same story was delivered in a conventional format using an illustrated book. In this regard, previous studies ([87]) have shown that children with ASD participating in computer-assisted reading sessions spend more time engaged with reading materials and demonstrated greater improvement in reading abilities, as well as less resistance than with the traditional book-based methods.

## 2. Materials and Methods

### 2.1. Participants

Following a convenience sampling method, consecutive individuals with ASD were recruited from the Neapolisanit Rehabilitation Centre in Naples, Italy, for rehabilitative purposes between February and July 2024. The Neapolisanit Centre is an accredited facility within the Italian national health system, specializing in education and rehabilitation for individuals with physical, psychological, and sensory disabilities. Its Autism Division (DAPI) is dedicated to supporting individuals with ASD and their families, aiming to enhance their quality of life and promote social inclusion. The center hosts a dedicated research division focused on autism. 

For all eligible participants, clinical records reported a suspected ASD diagnosis with varying levels of severity, supported by the previous administration of the Autism Diagnostic Observation Schedule, Second Edition (ADOS-2) ([51]). This is a standardized assessment tool widely used in ASD diagnostics. Like many other psychometric instruments, the ADOS-2 requires advanced examiner training. Furthermore, its sensitivity may vary across different clinical settings ([50]). While considered a valid and reliable tool, it is most effective when used in conjunction with other forms of assessment and within the context of clinical observation ([35]). Therefore, in the absence of a formal diagnosis made by an expert child neuropsychiatrist, participants were excluded from this study. Additional exclusion criteria were non-adherence to treatment, excessive concurrent training demands, and limited parental availability. Participants were stratified by age and ASD symptom severity before being randomly assigned to two groups (i.e., experimental group and control group) through a double-blind randomization procedure.

### 2.2. Apparatus and Stimuli

In this study, a story titled “*A Summer in the Garden*” was presented. The narrative followed the gardening tasks of the main character, “Pippo”, who seeks help planting a strawberry seedling at his grandmother’s country house. The story could be delivered orally by the examiner, supported by a booklet containing 14 illustrated pages, or through TUIs.

TUIs for digital storytelling are a methodology that leverages digital tools and technologies to convey stories, combining multimedia elements such as images, audio, videos, and objects to create engaging narratives. The digital version of the story, including 14 scenarios, was developed using the same 14 illustrations from the booklet, accompanied by audio narration. Additionally, participants were exposed to tangible objects or olfactory stimuli while listening to the story. The digital story was implemented using the STELT software (Smart Technologies to Enhance Learning and Teaching) ([58]). In its digital version, the story was presented by means of a standard laptop with a Windows OS (Figure 1a) and two Near Field Communication (NFC) antennas (Figure 1b; ([23])). NFC technology facilitates interaction with real-world objects. This setup used two types of NFC devices: an NFC Active Antenna, which acted as an interface between the computer and the physical objects, and NFC Tags, functioning as Passive Antennas. The NFC tags embedded in tangible objects held information readable by the active antenna. When the tag approached the active antenna, the antenna emitted an electromagnetic field, allowing the recognition of the corresponding object. Various tangible objects (e.g., a butterfly, watering can, shovel, soil sack, vase, strawberry seeds, and strawberry candy) and scent containers with fragrances like rose, pine, strawberry, and damp soil were incorporated into the story, creating a multisensory experience by linking the tactile and olfactory sensations of the objects and scents with the story’s audio-visual elements (Figure 1d). To enhance participants’ engagement, a 3D-printed figure of the story’s main character, “Pippo”, was also included, allowing participants to interact more closely with the protagonist (Figure 1c). 

In designing the apparatus for children with ASD, we adhered to some previously proposed guidelines ([74]):

*Size of tangible objects*: Tangible objects were designed to be appropriately sized, ensuring ease of grasping for children with limited fine motor skills.

*Size of the board containing the NFC antenna*: The board was sized to remain fully within the child’s arm’s reach, minimizing unnecessary movements and reducing potential distractions while ensuring comfort during the interaction.

*Sound usage*: Loud or abrupt sounds, such as buzzers or explosions, were avoided to accommodate the heightened auditory sensitivity often observed in children with ASD. Any warning sounds (e.g., to indicate an error; see Section 2.4) were emitted with soft tones.

### 2.3. Neuropsychological Testing

On admission, all participants were first screened through the ADOS-2, which is considered the reference standard assessment tool for supporting ASD diagnosis. This battery includes five different assessments: language and communication, mutual social interaction, imagination and creativity, stereotyped behavior and narrow interests, and other abnormal behaviors. For each domain, the appropriate module was selected based on the individual’s language level and age: *Module 1* was used for children who did not consistently use verbal language, *Module 2* for those who used phrase speech but were not yet fluent, and *Module 3* for children and adolescents who spoke fluently. The calibrated severity score (CSS) was employed to standardize the raw ADOS-2 scores, ensuring that ASD symptom severity could be meaningfully compared both across individuals with different developmental levels and over time. The CSS scale ranges from 1 to 10: a score between 1 and 3 indicates the presence of low levels of ASD symptoms (i.e., high functionality); a score between 4 and 5 indicates the presence of mild-to-moderate ASD symptoms (i.e., moderate functionality); and a score between 6 and 10 indicates the presence of moderate-to-severe symptoms (i.e., low functionality) ([72]).

Subsequently, specific subtests from the NEPSY-II ([16]; [42]), a neuropsychological battery for preschool and school-age subjects (from 3 to 16 years old), were administered. Overall, the NEPSY-II encompasses five domains: (1) Attention and Executive Functioning, (2) Language, (3) Memory and Learning, (4) Sensorimotor Functioning, and (5) Visuospatial Processing. Its broad range of subtests offers flexibility, allowing the examiner to select tasks best suited to their research aims ([16]). For this study, we chose subtests evaluating the child’s ability to follow instructions (Language domain) and perform properly episodic verbal memory tasks (Memory and Learning domain). For each administered test, the environment was tailored to meet the specific requirements of the task. Only the materials required to complete each subtest were provided, with all other items removed from the experimental setting to minimize distractions. Also, such an arrangement aimed to reduce anxiety in the child, which might otherwise concern about their ability to successfully complete the task. Below is a brief description of each subtest employed in this study.

*L1-Comprehension of Instructions (L1-C)*. This subtest is designed to assess the ability to perceive, process, and follow verbal instructions of increasing syntactic complexity and execute the corresponding actions. Specifically, the examinee is required to point to appropriate stimuli in response to oral instructions. The task materials consist of two pictorial stimuli: the first image features eight rabbits varying in color and size, while the second image displays geometric shapes differing in color and shape. After confirming the examinee’s ability to understand simple instructions (e.g., “touch mouth”, “touch head”), the stimulus images are presented, and instructions of progressively greater complexity are given based on the child’s age. The L1-C task can be administered to individuals aged 3 to 16 years. Its scoring range is 0 to 33, with higher scores indicating greater linguistic comprehension abilities.

*M4-List Memory (M4-LM) and M4-List Memory Delayed (M4-LMD).* These subtests are designed to assess both immediate and delayed recall of a supraspan word list, the rate of information acquisition, and the impact of proactive interference. In M4-LM, the examiner reads aloud a list of 15 words, and the participant is required to repeat them, recalling the words in any order. There are five learning trials, each followed by an immediate recall. The total number of words correctly recalled across the five trials constitutes the M4-LM raw score. After the fifth trial, the examiner presents a new list of 15 words, which the participant is also required to recall. Following this interference trial, a delay of 25–35 min occurs, during which the participant is not engaged in memory tasks. After the delay, the examinee is asked to recall all the words they remember from the original (first) list. The total number of words correctly recalled from the first list constitutes the M4-LMD raw score. These tasks can be administered to children aged 7 to 12 years. For M4-LM, the total score ranges from 0 to 75, while for M4-LMD, it ranges from 0 to 15. Higher scores reflect better episodic memory skills.

*M6-Narrative Memory Free Recall (M6-FR) and M6-Narrative Memory Cued Recall (M6-CR)*. These subtests assess the ability to recall verbally organized material under conditions of free recall and cued recall. The task begins with the examiner reading a story aloud to the child. They are asked to repeat everything they can remember from the story (M6-FR). Then, the examiner poses specific questions about the story to prompt recall of story-related information (M6-CR). Three different stories are available, selected based on the child’s age. The total M6-FR score ranges from 0 to 20, with one point awarded for each correctly recalled detail. Similarly, in M6-CR, the examiner asks 20 structured questions targeting specific story details, with one point assigned for each accurate response. Overall, higher scores suggest better episodic memory functioning.

The clinical interpretation of NEPSY-II scores follows the framework of the standard scaled score (SS), which represents a child’s performance relative to age-matched peers. The SS framework has a hierarchical structure: SS range 1–3 (Well below expected level), SS range 4–5 (Below expected level), SS range 6–7 (Borderline), SS range 8–12 (At expected level), and SS range 13–19 (Above expected level). In this study, performance improvement was operationally defined as a transition from a lower SS range to a higher SS range. For instance, an increase from SS 2 to SS 3 would not be classified as a significant improvement, as both scores fall within the same SS range. Conversely, a change from SS 6 to SS 10 would be considered clinically meaningful, reflecting a transition from the “Borderline” range to the “At expected level” range.

### 2.4. Procedure

All participants included in this study underwent two training sessions per week for four weeks, each session lasting one hour. At the beginning of the first session, all children underwent neuropsychological evaluation through the selected NEPSY-II subtests. In the experimental group, TUI-mediated digital storytelling was used. Each participant was asked to interact with objects while listening to the story. Some of these objects were elements of the story, while some others acted as distractors. If the child placed the right object on the keyboard, the story proceeded into the next scene; on the contrary, if it was the wrong object, the story stopped, and a mellow warning sound was emitted by the computer. As previously reported, all objects provide the subject with multisensory stimulation as they can be seen and touched, and some of them can be smelled. In the control group, children were asked to listen to the story read by the examiner while following it in the illustrated booklet. At the end of the treatment, the three NEPSY-II cognitive tests were re-administered to each participant. 

### 2.5. Ethical Approval

The data analyzed in the current study conformed with the Helsinki Declaration. Our named Ethical Committee of the Division for Educational Methodologies and Technologies for Autism of the Neapolisanit Rehabilitation Centre in Naples, Italy, specifically approved this study (protocol number IEC07/2023, February 2023). Prior to participation, the parents of the participants signed informed consent forms.

### 2.6. Statistical Analysis

Descriptive statistics were expressed as frequency for categorical variables and mean ± standard deviation (SD) for quantitative variables. Following verification of the relevant assumptions (e.g., normal distribution of residuals, homoscedasticity), between-group comparisons were performed using parametric (independent samples Student’s *t*-test) or non-parametric tests (e.g., 2-way χ^2^ test, Mann–Whitney U test) to determine whether the experimental and control groups were matched for sociodemographic variables, ADOS-2-related functioning level, and baseline (pre-test) neuropsychological test performance, as appropriate. An Aligned Rank Transform Linear Mixed Model (ART LMM) was constructed to investigate pre-post treatment differences in raw scores for each neuropsychological testing session. Specifically, the ART procedure involved preprocessing raw scores by removing all effects from the dependent variable except for the one being aligned. The aligned data were then ranked, with average ranks assigned in cases of ties. This approach ensures appropriate Type I error rates and adequate statistical power for main effects and interactions ([89]). Subsequently, an LMM was applied as it is more appropriate for handling repeated measures data compared with traditional Analysis of Variance ([18]). The LMMs included fixed effects for the group (experimental vs. control) and time (pre-test and post-test), while participants’ ID entered the model as a random effect ([39]). The Satterthwaite method was used to approximate degrees of freedom (*df*) for the fixed effects. Variance components of the random effects were estimated using Restricted Maximum Likelihood (REML). Post hoc pairwise comparisons were conducted using the ART-C procedure ([27]). Listwise deletion was applied to manage any missing values. Bonferroni’s adjustment was used for multiple comparisons, as appropriate. To further assess the potential benefit of the treatment, the McNemar test was applied to compare the number of participants who, following the treatment, achieved an SS within a higher score range. This non-parametric test is well suited for paired nominal data ([56]). Performance improvement was operationalized as a binary (dummy) variable, where 0 represented no improvement, and 1 represented an improvement. Effect sizes were estimated using R-squared statistics for LMMs, Cohen’s *d* for post hoc analyses, and psi (ψ) for the McNemar test ([19]; [26]; [46]). The nominal α level was 0.05. Aligned and ranked responses were generated using ARTool.exe version 2.2.2. LMMS were run using the *GAMLj* suite for Jamovi. Overall, statistical analyses were conducted using Jamovi v. 2.4.11. 

## 3. Results

### 3.1. Descriptive Statistics

Fifty children were enrolled. According to the exclusion criteria, the final cohort consisted of 28 participants. The children were randomly assigned to the experimental or control group, each comprising 14 participants. Descriptive statistics for demographic, clinical, and baseline neuropsychological characteristics of the two groups are reported in Table 1.

All participants were males, aged 4 to 13 years, with education levels ranging from 4 to 10 years, starting from kindergarten, according to the Italian schooling system. The ADOS-2 scores revealed a range of different functioning skills across the two groups, reflecting varying degrees of ASD severity: overall, six children demonstrated high functionality, nine moderate functionality, and 13 low functionality. The two groups were matched for age (t = −0.153, *df* = 26, *p* = 0.88), education level (t = −0.233, *df* = 26, *p* = 0.82), and ASD symptom severity (χ^2^ = 0.188, *df* = 2, *p* = 0.91). To ensure the interpretability of any findings, we verified that the experimental and control groups were comparable at the pre-test stage. The effectiveness of the randomization procedure was confirmed, as the two groups showed no significant difference in any administered NEPSY-II test scores (all *p* > 0.05).

### 3.2. Results from ART LMM Analysis

Here is a summary of the results for each NEPSY-II test, highlighting any main effects (group and/or time) and interactions (group × time). Table 2 shows the raw scores achieved by each participant included in this study. The results of ART LMM analyses are graphically represented in Figure 2. 

*L1-Comprehension of Instructions.* Both a main effect of time (*R*^2^ = 0.63, *F*_1,26_ = 4.867, *p* = 0.036) and a group × time interaction emerged (*R*^2^ = 0.72, *F*_1,26_ = 6.912, *p* = 0.014), while no main effect of group was observed (*F*_1,26_ = 1.006, *p* = 0.325). As for the main effect of time, post hoc analyses revealed a significant improvement in performance from pre- to post-treatment (t = 2.21, *df* = 26, *p_Bonferroni_* = 0.036, *d* = 0.87). This improvement is clearly attributable to the experimental group, although results from post hoc contrasts highlighted only a trend towards significance (experimental group, pre-test vs. post-test: t = −2.604, *df* = 25, *p_Bonferroni_* = 0.09, *d* = 0.87; control group, pre-test vs. post-test: t = −0.035, *df* = 25, *p_Bonferroni_* > 0.99). 

*M4-List Memory.* Both the main effect of group (*F*_1,21_ = 4.313, *p* = 0.05) and time (*F*_1,21_ = 3.932, *p* = 0.06) were found to border the nominal statistical level, with better overall performance in the experimental group (t = 2.08, *df* = 21, *p_Bonferroni_* = 0.05) and at the end of treatment (t = 1.98, *df* = 21, *p_Bonferroni_* = 0.06), respectively. No significant group × time interaction effect was detected (*F*_1,21_ = 1.77, *p* = 0.20). 

*M4-List Memory Delayed.* A statistically significant main effect of the group was highlighted (*R*^2^ = 0.40, *F*_1,21_ = 9.781, *p* = 0.005). Specifically, the experimental group outperformed the control group in this task (t = 3.13, *df* = 21, *p_Bonferroni_* = 0.005, *d* = 1.37). Also, a main effect of time (*F*_1,21_ = 3.845, *p* = 0.06) and a group × time interaction effect (*F*_1,21_ = 2.99, *p* = 0.09) were found to approach statistical significance. A trend toward improvement at post-test was found in the experimental group compared with the control group (t = 2.771, *df* = 20, *p_Bonferroni_* = 0.07). 

*M6-Narrative Memory-Free Recall.* No main effect of group (*F*_1,25_ = 0.780, *p* = 0.39) or time (*F*_1,25_ = 2.949, *p* = 0.10) nor any time interaction effect (*F*_1,25_ = 0.913, *p* = 0.35) emerged from the model.

*M6-Narrative Memory-Cued Recall.* The analysis did not reveal significant main effects for the group (*F*_1,25_ = 1.49, *p* = 0.23) or time (*F*_1,25_ = 0.187, *p* = 0.67) nor a significant group × time interaction (*F*_1,25_ = 0.696, *p* = 0.41). 

### 3.3. Proportion of Improvement in NEPSY-II

Given the challenges of fitting a semi-parametric approach with a limited sample size, McNemar’s test was used to compare the number of individuals in each group who improved their performance based on SS ranges. The results of McNemar’s test are depicted in Figure 3. As for L1-C, a significant treatment effect was observed, with 7/14 experimental participants improving at the post-test, while none improved in the control group (χ^2^ = 7.00, *df* = 1, *p* = 0.008, ψ = ∞). While M4-LM SS scores showed equal improvement across groups, with two participants improving in each group (χ^2^ = 5.33, *df* = 1, *p* = 0.021, ψ = 1), a moderate effect in favor of the experimental group was observed in M4-LMD (χ^2^ = 7.36, *df* = 1, *p* = 0.007, ψ = 2). In particular, two participants included in the experimental group improved at post-test compared with only one in the control group. Similarly, while no association between group and improvement was observed for M6-FR (χ^2^ = 3.00, *df* = 1, *p* = 0.08), five participants in the experimental group improved in M6-CR SS scores compared with only one in the control group (χ^2^ = 5.44, *df* = 1, *p* = 0.02, ψ = 5).

Given the small sample size, only a few participants are represented at each ASD severity level based on the ADOS-2 score. This makes it challenging to evaluate intervention efficacy based on aggregated data. However, analyzing individual-level outcomes may provide insight, especially in tasks where the proportion of experimental participants showing improvement was clinically meaningful. For instance, in L1-C, the proportion of children showing improvement was 33.3% (2/6) in the low-functioning group, 60% (3/5) in the moderate-functioning group, and 66.7% (2/3) in the high-functioning group (χ^2^ = 1.20, *df* = 2, *p* = 0.55). In M6-CR, improvement rates were 50% (3/6), 20% (1/5), and 50% (1/2), respectively (χ^2^ = 1.10, *df* = 2, *p* = 0.56). These preliminary findings suggest that the TUI-based intervention may effectively target language and memory deficits in ASD, regardless of the child’s level of functioning.

## 4. Discussion

This study aimed to investigate whether the use of a multisensory integration (MSI), combined with Tangible User Interfaces (TUIs) during a digital storytelling task, could improve comprehension and memory skills in a group of children with autism spectrum disorder (ASD). To test this hypothesis, the performance of a group of children using TUIs (experimental group) was compared to that of children who followed the story through corresponding images in a book (control group). Although the results are preliminary due to the small sample size, our findings appear promising. 

At the end of the intervention, the experimental group demonstrated a trend of improvement in tasks assessing instruction comprehension and narrative memory (cued recall), with a significantly greater proportion of children benefiting compared with the control group. Interestingly, the benefits of TUIs appeared to be evenly distributed across low-, moderate-, and high-functioning children, which suggests that their effectiveness is not contingent on the baseline severity of ASD symptoms. This further supports the potential of TUIs as a scalable and adaptable tool for intervention, capable of addressing core deficits in ASD across a wide range of functional profiles. According to the previous literature, these findings further highlight the potential of MSI for cognitive enhancement in neurodevelopmental disorders, including ASD ([25]). Consistent with the “redundant-target” paradigm ([11]; [15]), individuals respond faster to multisensory stimuli (e.g., sound and light combined) than to unisensory ones due to the facilitatory effects of MSI ([76]). 

Our results are strictly behavioral and do not allow conclusions about the neural substrates involved in the proposed treatment. However, one might hypothesize that regions responsible for early sensory processing and integrating auditory and visual information may have played a key role. Indeed, the primary task required audio-visual integration—listening to the story while viewing images (either on a screen or in a book) ([66]). Accordingly, several studies have shown that audio-visual integration amplifies activity in critical brain regions, such as the primary auditory cortex ([13]; [14]), visual motion and prefrontal regions ([64]; [90]), including Broca’s area, and the premotor cortex ([38]; [57]; [60]). Thus, it is plausible that these areas were similarly engaged during the intervention.

Interestingly, while children in both the experimental and control groups showed similar improvements on the free recall memory task, a significantly higher number of children in the experimental group improved on the cued narrative memory task. On the one hand, these findings support the effectiveness of both the experimental and control interventions in improving memory skills. On the other hand, the observed dissociation aligns with previous research ([8]; [10]; [88]) showing that while children were unable to recall the story in full, they could still answer specific questions about it. This suggests that the mnemonic difficulties observed in ASD children do not stem from an inability to encode the information but rather from retrieving it when no facilitation is provided. In other words, the issue may not pertain to the encoding of the memory trace but rather to the effective deployment of retrieval (executive) strategies. From an anatomic-functional standpoint, it has been observed that some areas involved in mnemonic encoding, such as the primary visual and auditory cortices, remain functionally intact in children with ASD, but access to stored information may depend on associative circuits that are less efficient ([93]). 

An innovative aspect of this study was the inclusion of olfactory stimulation alongside tactile, visual, and auditory inputs. Olfaction is unique among sensory modalities as it bypasses the thalamus and directly accesses limbic structures such as the hippocampus and the entorhinal cortex, which are critical for memory and spatial navigation ([94]). Notably, the entorhinal cortex contains place cells, neurons that play a central role in encoding spatial and contextual information ([70]). This direct link between olfactory processing and the hippocampal memory system suggests that in children with ASD, who often exhibit deficits in encoding and retrieving episodic memories, olfactory inputs might have supported the encoding of the narrative’s structure, aiding retrieval during the cued recall task ([8]; [10]; [88]). This hypothesis aligns with findings on neurotypical individuals, where odors were found to evoke vivid and emotionally rich memories ([36]). Indeed, although data on olfactory function in ASD are still controversial ([9]; [20]; [68]; [86]), it has been shown that relative to typically developing controls, individuals with ASD demonstrated increased functional connectivity of the olfactory network and decreased functional connectivity within the multisensory integration network ([6]; [71]). As the thalamus acts as a relay center for all sensory modalities except olfaction and plays a key role in facilitating MSI during combined stimulation ([33]; [82]), we might hypothesize that, in our study, olfactory stimulation may have facilitated the creation of stronger intermodal connections. As a result, this may have enhanced the benefits of multisensory stimulation ([53]; [69]; [77]). A recent review ([4]) has argued in favor of studying olfactory stimulation, as well as visuo-olfactory integration, given the potential of these approaches to drive effective interventions in ASD.

Beyond clinical perspectives, the potential of TUIs within educational settings should not be overlooked ([22]; [23]). Italy boasts a long-standing tradition of inclusive education with decades of legislation aimed at integrating students with disabilities into mainstream schools. Since the 1970s, this framework has led to the elimination of segregated classrooms, the introduction of specialist support teachers, and the implementation of a collaborative model that connects schools, families, and local support services. This ensured that students received coordinated assistance across different settings, benefiting from a holistic approach to education. It is important to stress that not all European countries have embraced a fully inclusive approach to education. In Germany and the Netherlands, for instance, students with disabilities are primarily registered in specialized schools. This policy contrasts with the documented benefits of full inclusion, which has been associated with reduced social isolation, improved interpersonal skills, and enhanced self-esteem and self-efficacy among students with emotional and/or behavioral disorders ([81]).

Although Italy has positioned itself at the forefront of inclusive education policies, conventional educational strategies often struggle to fully engage students with ASD. Standard teaching models may not fully address the diverse sensory and cognitive needs within this population, underscoring the need for innovative tools that support individualized, engaging, and accessible learning experiences within inclusive settings. In this vein, TUIs may offer a structured yet flexible solution. By leveraging physical interaction and minimizing reliance on abstract, verbal-based instruction, TUIs may support an alternative, cutting-edge learning paradigm. For instance, their ability to facilitate engagement through hands-on, multisensory learning experiences, combined with their capacity to foster social interaction—whether with the examiner or within peer groups in cooperative activities—makes them particularly well-suited for inclusive educational environments designed to support individuals with ASD.

The present study is not without limitations. One key constraint is the limited sample size, which restricts the generalizability of our findings. Also, this study lacks a control group of typically developing children, making it difficult to determine whether the observed benefits of the multisensory intervention are specific to children with ASD or reflect broader trends across the general population. Moreover, as a result of the convenience sampling procedure, our study includes only boys. While autism is about four times more prevalent in males than females ([52]), making this focus justified, our sample still does not fully capture the epidemiological distribution of the phenomenon. Further studies with balanced groups are needed to explore sex-related differences in autism presentation and to test the invariance of TUIs’ effectiveness as an intervention paradigm.

## 5. Conclusions

This study suggests that a TUI-mediate multisensory intervention holds promise for enhancing cognitive abilities in children with ASD. This approach could not only improve comprehension and memory but also pave the way for innovative treatments that leverage brain plasticity and functionally preserved areas. Future research should seek to validate these results with larger samples, including neurophysiological measures and advanced neuroimaging techniques for investigating the neural mechanisms involved.

## Figures and Tables

**Figure 1 behavsci-15-00267-f001:**
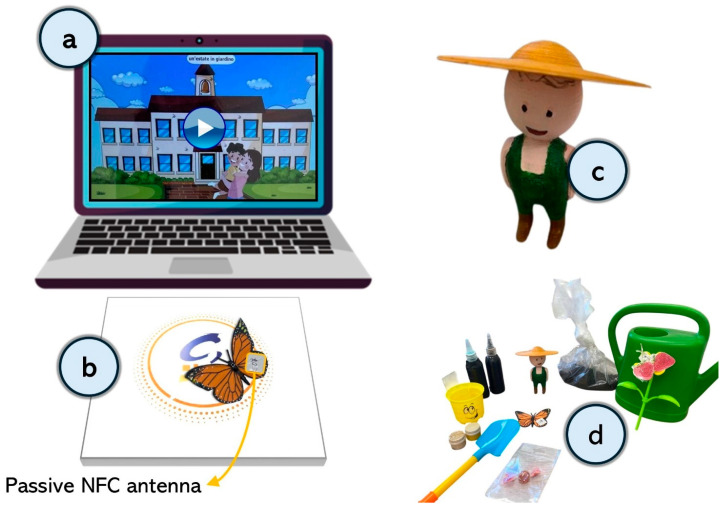
Tangible User Interface (TUI) apparatus and multisensory stimuli for digital storytelling. The TUI prototype consisted of (**a**) a system with a laptop running Windows OS to play the video and (**b**) a rigid plastic board, called the “Magic board”, connected via USB cable. The Magic board contained an active Near Field Communication (NFC) antenna. Each object used in this study was equipped with a tag embedded with a passive NFC antenna. When an object interacted with the Magic board, the system identified it. If the selected object was correct, the story progressed; otherwise, the story paused, and the computer emitted a warning sound. Among the interactive objects was “Pippo”, the story’s protagonist, represented as a 3D-printed figurine (**c**). During the task, the selection of objects was guided by their multisensory properties (**d**). For touch, these included a small hand rake, soil, a watering can, a strawberry plant, and Pippo. The smell was engaged through containers emitting scents such as pine, rose, strawberry, and damp earth. Finally, the taste was stimulated with a strawberry-flavored candy.

**Figure 2 behavsci-15-00267-f002:**
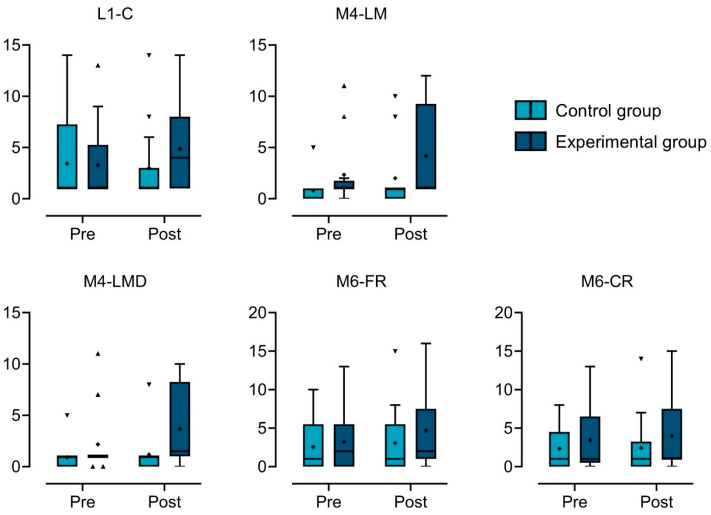
Box-and-whisker plots illustrating the performance (raw scores) at pre- and post-test of the experimental and control groups across the five NEPSY-II subtests. Tukey’s method was used to plot whiskers and outliers. The ends of the boxes represent the first (Q1) and third (Q3) quartiles, with the interquartile range (IQR) defined as the difference between Q3 and Q1. Values greater than Q3 plus 1.5 times the IQR or less than Q1 minus 1.5 times the IQR are plotted as individual points. The median is displayed as a horizontal line within the box, while the arithmetical mean is marked with a “+” symbol. *Note:* L1-C: L1-Comprehension of Instructions; M4-LM: M4-List Memory; M4-LMD: M4-List Memory Delayed; M6-FR: M6-Narrative Memory-Free Recall; M6-CR: M6-Narrative Memory-Cued Recall.

**Figure 3 behavsci-15-00267-f003:**
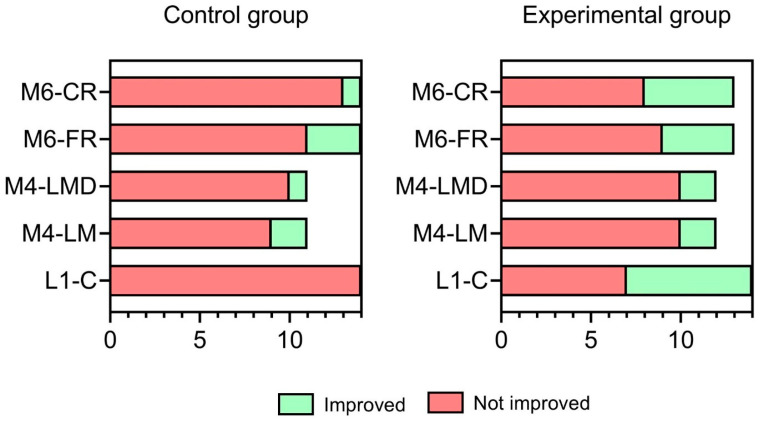
Stacked bar charts depicting the proportions of participants who showed improvement in each of the five NEPSI-II subtests following the treatment. *Note:* L1-C: L1-Comprehension of Instructions; M4-LM: M4-List Memory; M4-LMD: M4-List Memory Delayed; M6-FR: M6-Narrative Memory-Free Recall; M6-CR: M6-Narrative Memory-Cued Recall.

**Table 1 behavsci-15-00267-t001:** Demographic, clinical, and baseline neuropsychological characteristics of the two groups.

	Experimental Group (*n* = 14)	Control Group (*n* = 14)	*p*-Value
Age, years; mean (SD)	9.14 (2.80)	9.00 (2.07)	0.88 ^a^
Education, year; mean (SD)	6.21 (2.75)	6.00 (2.07)	0.82 ^a^
*Functional level (ADOS-2)*			
Low functioning, CSS range: 6–10	6	7	
Moderate functioning, CSS range: 4–5	5	4	
High functioning, CSS range: 1–3	3	3	
L1-C, raw score; mean (SD)	3.29 (3.77)	3.43 (4.13)	0.92 ^a^
M4-LM, raw score; mean (SD)	2.33 (3.45) ^†^	0.82 (1.47) ^††^	0.19 ^a^
M4-LMD, raw score; mean (SD)	2.17 (3.33) ^†^	0.91 (1.45) ^††^	0.21 ^b^
M6-FR, raw score; mean (SD)	3.23 (3.81) ^⁋^	2.57 (3.39)	0.53 ^a^
M6-CR, raw score; mean (SD)	3.46 (3.93) ^⁋^	2.36 (2.90)	0.35 ^a^

*Note:* CSS: Calibrated Severity Scores; L1-C: L1-Comprehension of Instructions; M4-LM: M4-List Memory; M4-LMD: M4-List Memory Delayed; M6-FR: M6-Narrative Memory-Free Recall; M6-CR: M6-Narrative Memory-Cued Recall. ^†^ *n* = 12. ^††^ *n* = 11. ^⁋^
*n* = 13. ^a^ Student’s *t*-test. ^b^ Mann–Whitney U test.

**Table 2 behavsci-15-00267-t002:** Number of correct responses recorded for each NEPSY-II subtest at pre- and post-test.

ID	Group	L1-C		M4-LM		M4-LMD		M6-FR		M6-CR	
		Pre	Post	Pre	Post	Pre	Post	Pre	Post	Pre	Post
P1	EG	1	1	1	3	1	2	1	1	1	1
P2	EG	1	2	1	1	1	2	3	1	3	2
P3	EG	13	14	11	11	11	10	13	15	13	15
P4	EG	4	5	•	•	•	•	5	5	7	10
P5	EG	1	1	1	1	1	1	0	0	0	1
P6	EG	1	1	0	1	0	0	0	1	0	1
P7	EG	1	6	1	1	1	1	2	2	1	1
P8	EG	5	6	1	12	1	10	7	8	7	9
P9	EG	1	1	0	1	0	1	0	0	0	0
P10	EG	6	8	2	1	1	1	4	4	5	6
P11	EG	1	3	1	10	1	9	1	7	1	4
P12	EG	9	11	8	7	7	6	6	16	6	1
P13	EG	1	8	•	•	•	•	•	•	•	•
P14	EG	1	1	1	1	1	1	0	1	1	1
P15	CG	14	14	0	0	1	0	7	15	6	14
P16	CG	1	1	1	1	0	1	1	1	1	1
P17	CG	2	2	•	•	•	•	1	1	1	3
P18	CG	1	1	0	1	0	1	0	0	0	0
P19	CG	1	2	•	•	•	•	5	7	4	4
P20	CG	1	1	0	1	1	1	0	1	0	0
P21	CG	1	2	1	0	0	0	1	1	2	2
P22	CG	7	6	•	•	•	•	7	8	8	1
P23	CG	8	1	5	10	0	8	4	4	4	2
P24	CG	1	1	0	0	5	0	0	0	0	0
P25	CG	8	8	1	8	1	1	10	5	7	7
P26	CG	1	1	0	0	1	0	0	0	0	0
P27	CG	1	1	0	0	0	0	0	0	0	0
P28	CG	1	1	1	1	1	1	0	0	0	0

*Note:* ID: Identification Code; EG: experimental group; CG: control group; L1-C: L1-Comprehension of Instructions; M4-LM: M4-List Memory; M4-LMD: M4-List Memory Delayed; M6-FR: M6-Narrative Memory-Free Recall; M6-CR: M6-Narrative Memory-Cued Recall. In line with the demographic guidelines of the NEPSI-II, participants for whom the task was deemed not applicable were labeled with “•”.

## Data Availability

The experimental data that support the findings of this study are available from the corresponding author upon reasonable request.

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
