# Peer review of "Using Tangible User Interfaces (TUIs): Preliminary Evidence on Memory and Comprehension Skills in Children with Autism Spectrum Disorder"

_behavsci, 2025, doi:10.3390/bs15030267_

Round 1

Reviewer 1 Report

Comments and Suggestions for Authors

Dear Authors,

I would like to express my gratitude for the opportunity to review your manuscript and extend my sincere congratulations for this thoughtful and rigorous work. Your inclusion of olfactory elements in the experimental protocol is innovative, and your commitment to presenting robust statistical analyses is particularly commendable. These efforts significantly enhance the methodological strength of your study.

While your manuscript represents a promising contribution to the field, I would like to offer several suggestions that could help improve its clarity, comprehensiveness, and relevance to a broader audience:

- The section addressing ABA is notably incomplete. It is essential to acknowledge the body of research highlighting the potential negative effects, methodological challenges, and conflicts of interest associated with ABA interventions. Including these perspectives would provide a more balanced view of this widely debated topic.

- For the benefit of readers unfamiliar with the Neapolisanit Center, it would be helpful to include additional information about this institution, its role in recruitment, and any associated potential recruitment biases. This transparency will strengthen the contextual framework of your study.

- While the ADOS-2 is a widely used tool, it is not without limitations. A discussion on the potential biases inherent in its application, particularly when diagnosing or assessing individuals across diverse populations, would add depth to your methodological considerations.

- One of the study’s notable limitations is the absence of a comparison group of typically developing (TD) children. Without this, it is difficult to determine whether the observed benefits of the multisensory approach are unique to children with ASD or if they reflect broader trends in learning across populations.

- Including the validation scores for the study populations in Table 1 would provide readers with a clearer understanding of the representativeness and reliability of your sample.

- The exclusive focus on boys in your sample warrants further discussion. This limitation should be explicitly addressed, considering the growing awareness of how autism spectrum disorder manifests differently in girls and boys.

- The manuscript would benefit from a brief discussion of the unique characteristics of Italy’s inclusive educational system, particularly in light of its long-standing legislation on specialized education. Comparing these features to other educational systems would provide valuable context for international readers.

- A more detailed discussion of individual-level outcomes, particularly in relation to the participants’ levels of functioning and specific needs, would help contextualize the findings and their practical implications.

- Certain terms, such as “treatment,” imply a medicalized view of autism as a condition to be cured. I encourage you to consider alternative terminology that aligns with the neurodiversity paradigm, which frames autism as a difference rather than a disease.

By addressing these points, I believe the manuscript would not only gain in clarity and rigor but also resonate more effectively with a broader, interdisciplinary audience. I look forward to seeing the revised version and commend you again for your contributions to this field.

Sincerely.

Author Response

Dear Authors, I would like to express my gratitude for the opportunity to review your manuscript and extend my sincere congratulations for this thoughtful and rigorous work. Your inclusion of olfactory elements in the experimental protocol is innovative, and your commitment to presenting robust statistical analyses is particularly commendable. These efforts significantly enhance the methodological strength of your study.

Reply: We sincerely appreciate the Reviewer’s thoughtful and encouraging feedback regarding our study’s methodology.

While your manuscript represents a promising contribution to the field, I would like to offer several suggestions that could help improve its clarity, comprehensiveness, and relevance to a broader audience:

The section addressing ABA is notably incomplete. It is essential to acknowledge the body of research highlighting the potential negative effects, methodological challenges, and conflicts of interest associated with ABA interventions. Including these perspectives would provide a more balanced view of this widely debated topic.

Reply: We fully agree with Reviewer 1. Accordingly, we have added a dedicated paragraph highlighting the limitations of the ABA paradigm, including its potential methodological weaknesses (see lines 80-105).

For the benefit of readers unfamiliar with the Neapolisanit Center, it would be helpful to include additional information about this institution, its role in recruitment, and any associated potential recruitment biases. This transparency will strengthen the contextual framework of your study.

Reply: We thank Reviewer 1 for this suggestion. We have added the relevant information about the Neapolisanit Centre in lines 198-205.

While the ADOS-2 is a widely used tool, it is not without limitations. A discussion on the potential biases inherent in its application, particularly when diagnosing or assessing individuals across diverse populations, would add depth to your methodological considerations.

Reply: We fully agree with Reviewer 1. In the revised version of our manuscript, we have added the well-known limitations of the ADOS-2. Additionally, we have clarified that only participants with a diagnosis of ASD confirmed by an expert child neuropsychiatrist were included. Conversely, those assessed solely by using the ADOS-2 were excluded. See lines 206–215.

One of the study’s notable limitations is the absence of a comparison group of typically developing (TD) children. Without this, it is difficult to determine whether the observed benefits of the multisensory approach are unique to children with ASD or if they reflect broader trends in learning across populations.

Reply: We thank Reviewer 1 for his/her valuable suggestion. The lack of a control group of typically developing children is indeed a limitation of our study, which we have now explicitly acknowledged at the end of the Discussion section (see lines 605–608).

Including the validation scores for the study populations in Table 1 would provide readers with a clearer understanding of the representativeness and reliability of your sample.

Reply: We thank the reviewer for his/her comment. In the revised version of the manuscript, Table 1 reports, for each ADOS-2 severity level, the number of participants in that category and the corresponding standardized score ranges based on the Calibrated Severity Scores (CSS).

The exclusive focus on boys in your sample warrants further discussion. This limitation should be explicitly addressed, considering the growing awareness of how autism spectrum disorder manifests differently in girls and boys.

Reply: We agree with Reviewer’1 criticism. However, recent epidemiological data report a higher prevalence of ASD in males than in females (Maenner, 2023). The focus on male participants is therefore justified, though primarily driven by convenience sampling. Nonetheless, we acknowledge the absence of female participants as a limitation, which we have emphasized at the end of the Discussion section (see lines 608–614).

The manuscript would benefit from a brief discussion of the unique characteristics of Italy’s inclusive educational system, particularly in light of its long-standing legislation on specialized education. Comparing these features to other educational systems would provide valuable context for international readers.

Reply: We greatly appreciated this suggestion and took the opportunity to expand the discussion on this topic. We not only have emphasized Italy’s leadership in inclusive education by briefly outlining its legislative history and comparing its policies to those of other countries, but we have also contextualized the discussion by highlighting how TUIs can complement traditional teaching methods. Given their significant potential for ASD interventions, TUIs represent a valuable tool for both educational and clinical applications (see lines 578-603):

A more detailed discussion of individual-level outcomes, particularly in relation to the participants’ levels of functioning and specific needs, would help contextualize the findings and their practical implications.

Reply: We sincerely thank Reviewer 1 for this valuable suggestion. Indeed, given our limited sample size, exploring the relationship between functioning level and intervention outcomes is likely only possible at individual-level. In this regard, we have highlighted that in the most relevant tasks—where the experimental group showed the greatest improvements compared to the control one (LC-1 and M6-CR)—there were no proportional differences across functioning levels (see lines 489–499). These preliminary observations allow us to suggest that TUIs may be effective as an intervention paradigm regardless of ASD symptom severity (see lines 517–521).

Certain terms, such as “treatment,” imply a medicalized view of autism as a condition to be cured. I encourage you to consider alternative terminology that aligns with the neurodiversity paradigm, which frames autism as a difference rather than a disease.

Reply: To align with the principles of neurodiversity, we have removed all references to “treatments” and replaced them with “intervention” throughout the manuscript.

By addressing these points, I believe the manuscript would not only gain in clarity and rigor but also resonate more effectively with a broader, interdisciplinary audience. I look forward to seeing the revised version and commend you again for your contributions to this field.

Reply: We have really appreciated Reviewer 1 remarks and constructive feedback. We trust that the revisions made adequately address his/her suggestions and enhance the clarity and rigor of our work.

Reviewer 2 Report

Comments and Suggestions for Authors
  1. The work reported is relevant, of a good quality, clearlty presented and written. The paper is well organized
  2. But, please leave completely aside your speculations regarding possible anatomo-physiological underpinnings of the data.
  3. It is well known in cognitive psychology that multisensory stimulation increases memory recall, as the more traces you have to recall from, the more chances you have to mobilize relevant associations in episodic memory stores.
  4. (Limited) multisensory integration has been shown to be beneficial for learning in people with ASD as you document in your review of the literature.
  5. The contribution of this research is that the use of a tangible user interface  (the way you have set it) yields better results than vision and sound alone.
  6. Fine, this adds another proven (likely better) technique for enhancing learning in ASD people. 
  7. It would have been useful for future intervention attempts in this field to compare systematically the relative efficiency of several intervention strategies according to number and type of user interface: vision alone, sound alone, vision + touch, vision + touch / TUI, and other possible patterns of stimulations, with separate and balenced sub-samples of participants., and perhaps be able on this basis to isolate particular relationships between pattern of stimulation and contents of the recalls. May be in a future work?
  8. Now a possibly puzzling question that i am raising to you and to the researchers in this field (no need of course to deal with it in this paper) : given that in real life ASD people like anyone of us are confronted with multiple stimulations (touch, smell, vision, sounds, etc.), and given as you demonstrate (as well as other reports) that they are indeed able to use multiple stimulation for learning in experimental settings, how is it that they don't do it in real life whereby concrete benefits and reinforcements hang around ?. What may be basically wrong or immature in their approach to the real life situations? (don't tell me that this is ADS, that would be a tautology!).
Comments on the Quality of English Language

See above

Author Response

The work reported is relevant, of a good quality, clearly presented and written. The paper is well organized.

Reply: We thank Reviewer 2 for his/her positive feedback.

But, please leave completely aside your speculations regarding possible anatomo-physiological underpinnings of the data.

Reply: We thank Reviewer 2 for sharing with us his/her concerns and perspective; however, the inclusion of speculations on potential neural correlates was explicitly requested by the Editor of the special issue to which we are submitting this manuscript.

It would have been useful for future intervention attempts in this field to compare systematically the relative efficiency of several intervention strategies according to number and type of user interface: vision alone, sound alone, vision + touch, vision + touch / TUI, and other possible patterns of stimulations, with separate and balanced sub-samples of participants., and perhaps be able on this basis to isolate particular relationships between pattern of stimulation and contents of the recalls. May be in a future work?

Reply: We thank Reviewer 2 for sharing with us this interesting comment. Quantifying the effectiveness of individual sensory channels and their interactions within the context of TUIs is indeed an objective we have considered. In future studies, we would like to explore this issue further in both healthy and clinical populations, including individuals in developmental stages as well as elderly people with acquired brain injuries.

-Now a possibly puzzling question that i am raising to you and to the researchers in this field (no need of course to deal with it in this paper) : given that in real life ASD people like anyone of us are confronted with multiple stimulations (touch, smell, vision, sounds, etc.), and given as you demonstrate (as well as other reports) that they are indeed able to use multiple stimulation for learning in experimental settings, how is it that they don't do it in real life whereby concrete benefits and reinforcements hang around ?. What may be basically wrong or immature in their approach to the real life situations? (don't tell me that this is ADS, that would be a tautology!).

Reply: This is a fascinating and complex question, touching on fundamental aspects of perception and cognition. While it is true that real-life environments provide a wealth of multisensory stimuli, the key difference between these naturalistic/ecological settings and controlled paradigms lies in the stimuli structure and predictability. In real-life scenarios, sensory information is abundant but unstructured. Multisensory input in everyday situations is dynamic, unpredictable, and often overwhelming, making it challenging, especially for individuals with ASD or similar populations, to extract meaningful patterns or leverage cross-modal integration effectively. Unlike in experimental/standardized settings, where tasks are designed to direct attention towards relevant stimuli, real-world environments do not inherently provide this architecture. This aligns with what is observed in other clinical populations—such as stroke survivors or individuals with dementia—who benefit from multisensory stimulation only when it is systematically delivered, rather than passively encountered in everyday life. Also, attentional control mechanisms play a crucial role. The ability to filter, prioritize, and integrate sensory input is task-dependent, meaning that the cognitive system selects and processes information according to situational demands. Individuals with ASD often exhibit deficits in attentional allocation and sensory filtering, which may limit their ability to spontaneously utilize multisensory integration in unstructured environments.

It is well known in cognitive psychology that multisensory stimulation increases memory recall, as the more traces you have to recall from, the more chances you have to mobilize relevant associations in episodic memory stores.

(Limited) multisensory integration has been shown to be beneficial for learning in people with ASD as you document in your review of the literature.

The contribution of this research is that the use of a tangible user interface  (the way you have set it) yields better results than vision and sound alone.

Fine, this adds another proven (likely better) technique for enhancing learning in ASD people. 

Reply: We really thank Reviewer 2 for his/her encouraging feedback.